# Improving Generalization in Reinforcement Learning with Mixture Regularization

**Kaixin Wang**[1]  **Bingyi Kang**[1]  **Jie Shao**[2]  **Jiashi Feng**[1]
[1]National University of Singapore    [2]ByteDance AI Lab
{kaixin.wang, kang}@u.nus.edu, shaojie.mail@bytedance.com, elefjia@nus.edu.sg

## Abstract

Deep reinforcement learning (RL) agents trained in a limited set of environments tend to suffer overfitting and fail to generalize to unseen testing environments. To improve their generalizability, data augmentation approaches (e.g. cutout and random convolution) are previously explored to increase the data diversity. However, we find these approaches only locally perturb the observations regardless of the training environments, showing limited effectiveness on enhancing the data diversity and the generalization performance. In this work, we introduce a simple approach, named *mixreg*, which trains agents on a mixture of observations from different training environments and imposes linearity constraints on the observation interpolations and the supervision (e.g. associated reward) interpolations. Mixreg increases the data diversity more effectively and helps learn smoother policies. We verify its effectiveness on improving generalization by conducting extensive experiments on the large-scale Procgen benchmark. Results show mixreg outperforms the well-established baselines on unseen testing environments by a large margin. Mixreg is simple, effective and general. It can be applied to both policy-based and value-based RL algorithms. Code is available at https://github.com/kaixin96/mixreg.

## 1 Introduction

Deep Reinforcement Learning (RL) has brought significant progress in learning policies to tackle various challenging tasks, such as board games like Go [19, 21], Chess and Shogi [20], video games like Atari [15, 1] and StarCraft [27], and robotics control tasks [14]. Despite its outstanding performance, deep RL agents tend to suffer poor generalization to unseen environments [31, 23, 29, 3, 2, 30]. For example, in video games, agents trained with a small set of levels struggle to make progress in unseen levels of the same game [2]; in robotics control, agents trained in simulation environments of low diversity generalize poorly to the realistic environments [26]. Such a generalization gap has become a major obstacle for deploying deep RL in real applications.

One of the main causes for this generalization gap is the limited diversity of training environments [29, 3, 2]. Motivated by this, some works propose to improve RL agents' generalizability by diversifying the training data via data augmentation techniques [3, 13, 12, 11]. However, these approaches merely augment the observations individually with image processing techniques, such as random crop, patch cutout [4] and random convolutions [13]. As shown in Figure 1 (left), such techniques are performing local perturbation within the state feature space, which only incrementally increases the training data diversity and thus leads to limited generalization performance gain. This is evidenced by our findings that these augmentation techniques fail to improve generalization performance of the RL agents when evaluated on a large-scale benchmark (see Section 4.1).

In this work, we introduce *mixreg* that trains the RL agent on a mixture of observations collected from different training environments. Inspired by the success of mixup [32] in supervised learning,

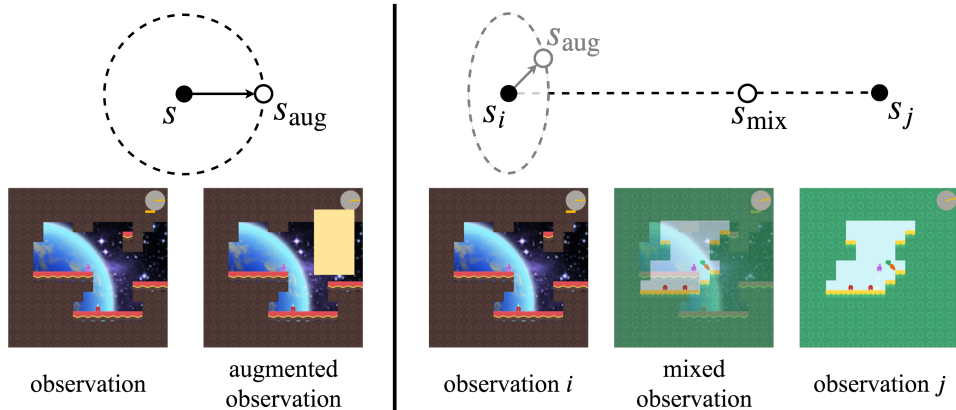

| observation | augmented observation | observation $i$ | mixed observation | observation $j$ |

Figure 1: **Left**: Previous data augmentation techniques (e.g. cutout) only apply local perturbations over the observation ($s \rightarrow s_{\text{aug}}$); they independently augment each observation regardless of training environments and achieve limited data diversity increment. **Right**: Our mixreg method smoothly interpolates observations from different training environments ($s_i, s_j \rightarrow s_{\text{mix}}$) thus producing more diverse data.

per training step, mixreg generates augmented observations by convexly combining two observations randomly sampled from the collected batch, and trains the RL agent on them with their interpolated supervision signal (e.g. the associated rewards or state values). In this way, the generated observations are widely distributed between the diverse observations and can effectively increase the training data diversity, as shown in Figure 1 (right). Moreover, mixreg imposes piece-wise linearity regularization to the learned policy and value functions w.r.t. the states. Such regularization encourages the agent to learn a smoother policy with better generalization performance. Notably, mixreg is a general scheme and can be applied to both policy-based and value-based RL algorithms.

We evaluate mixreg on the recently introduced Procgen Benchmark [2]. We compare mixreg with three best-performing data augmentation techniques (i.e. cutout-color, random crop, random convolution) in [12], and two regularization techniques (i.e. batch normalization [10] and $\ell_2$ regularization) adopted in previous works [3, 6]. We find that mixreg boosts the generalization performance of the RL agent more significantly, surpassing the baselines by a large margin. Moreover, when combined with other methods such as $\ell_2$ regularization, mixreg brings further improvement. We also verify the effectiveness of mixreg for both policy-based and value-based algorithms. Additionally, we conduct several analytical experiments to study and provide better understanding on its effectiveness.

This work makes the following contributions.

- We are among the first to study how to effectively increase training data diversity to improve RL generalization. Different from data augmentation techniques as commonly adopted in recent works, we propose to look into mixing observations from different environments.

- We introduce mixreg, a simple and effective approach for improving RL generalization by learning smooth policy over mixed observations. Mixreg can be easily deployed for both policy and value-based RL algorithms.

- On the recent large-scale Procgen benchmark, mixreg outperforms many well-established baselines by large margins. It also serves as a strong baseline for future studies.

## 2 Background

**Reinforcement learning** We denote an RL task (usually corresponding to an environment) as $\mathcal{K} = (\mathcal{M}, \mathcal{P}_0)$ where $\mathcal{M} = (\mathcal{S}, \mathcal{A}, \mathcal{P}, R)$ is a Markov Decision Process (MDP) with state space $\mathcal{S}$, action space $\mathcal{A}$, transition probability function $\mathcal{P}$ and the immediate reward function $R$. $\mathcal{P}(s, a, s')$ denotes the probability of transferring from state $s$ to $s'$ after action $a$ is taken, while $\mathcal{P}_0$ represents the distribution on the initial states $\mathcal{S}_0 \subset \mathcal{S}$. A policy is defined as a mapping $\pi : \mathcal{S} \rightarrow \mathcal{A}$ that returns an action $a$ given a state $s$. The goal of RL is to find an optimal policy $\pi^*$ which maximizes the

expected cumulative reward:

$$\pi^* = \arg\max_{\pi \in \Pi} \mathbb{E}_{\tau \sim \mathcal{D}_\pi} \sum_{t=0}^{T} \gamma^t R_t, \tag{1}$$

where $\Pi$ is the set of policies, $\tau$ denotes a trajectory $(s_0, a_0, s_1, a_1, \ldots, s_T)$, $\gamma \in (0, 1]$ is the discount factor, and $\mathcal{D}_\pi$ denotes the distribution of $\tau$ under policy $\pi$. RL algorithms can be categorized into policy-based and value-based ones, which will be briefly reviewed in the following. In Section 3, we will present how to augment them with our proposed mixreg.

**Policy gradient**  Policy gradient methods maximizes the objective in Eqn. (1) by directly conducting gradient ascent w.r.t. the policy based on the estimated policy gradient [25]. In particular, at each update, policy gradient maximizes the following surrogate objective, whose gradient is the policy gradient estimator:

$$L^{\text{PG}}(\theta) = \hat{\mathbb{E}}_t \left[\log \pi_\theta(a_t | s_t) A_t\right], \tag{2}$$

where $A_t$ is the estimated advantage function at timestep $t$, $\theta$ denotes the trainable parameters of the policy. Here $\hat{\mathbb{E}}_t[\cdot]$ denotes the empirical average over a collected batch of transitions. A learned state-value function $V(s)$ is often used to reduce the variance of advantage estimation. In this work, we use Proximal Policy Optimization (PPO) [18] for its strong performance and direct comparison with previous works. Details about PPO are given in the supplementary material.

**Deep Q-learning**  Deep Q-learning methods approximate the optimal policy by first learning an estimate of the expected discounted return (or value function) and then constructing the policy from the learned value function [15]. More specifically, at each update, Q-learning minimizes the following loss function

$$L^{\text{DQN}}(\theta) = \hat{\mathbb{E}}_t \left[ \left( R_t + \gamma \max_{a'} Q_{\bar{\theta}}(s'_t, a') - Q_\theta(s_t, a_t) \right)^2 \right], \tag{3}$$

where $Q$ represents the state-action value function with learnable parameters $\theta$. $\bar{\theta}$ denotes network parameters used to compute the value target. Following [2], we use a Deep Q-Network (DQN) variant Rainbow [8], which combines six extensions of the DQN algorithm. Details about Rainbow can be found in the supplementary material.

## 3  Method

### 3.1  Generalization in RL

To assess the generalization ability of an RL agent, we consider a distribution of environments $p(\mathcal{K})$. The agent is trained on a fixed set of $n$ environments $\mathcal{K}_{\text{train}} = \{\mathcal{K}_1, \cdots, \mathcal{K}_n\}$ (e.g. $n$ different levels of a video game) with $\mathcal{K}_i \sim p(\mathcal{K})$ and then tested on environments drawn from $p(\mathcal{K})$. Following [2], we use agents' zero-shot performance on testing environments to measure the generalization:

$$\mathbb{E}_{\tau \sim \mathcal{D}_{\hat{\pi}}^{\text{test}}} \sum_{t=0}^{T} \gamma^t R_t \tag{4}$$

where $\hat{\pi}$ is the policy learned on training environments while $\mathcal{D}_{\hat{\pi}}^{\text{test}}$ denotes the distribution of $\tau$ from the testing environments. The above performance depends on the difference between training and testing environments, which is the main cause of generalization gap. When $n$ is small, the training data diversity is also small and cannot fully represent the whole distribution $p$, leading to large training-testing difference. Consequently, the trained agent tends to overfit to the training environments and yield poor performance on the testing environments, showing large generalization gap. The difference may come from the environment visual changes [2, 3, 7, 30], dynamical changes [16] or structural changes [28]. In this work, we focus on tackling the visual changes. However, the proposed method is general and can be applied for other kinds of changes.

### 3.2  Mixture regularization

Inspired by the success of mixup in supervised learning [32], we introduce mixture regularization (*mixreg*) to increase the diversity of limited training data and thus minimize the generalization gap.

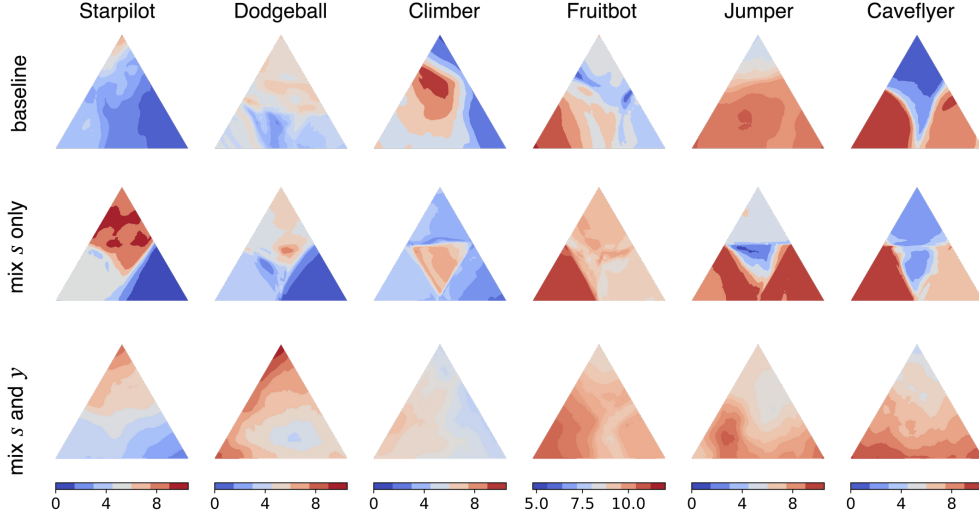

Figure 2: Visualization of the learned state-value functions $V(s)$ in PPO for 6 game environments from Procgen (from left to right). For each game, we plot the value predictions $V(s)$ for observation $s$ within the convex hull of three randomly selected observations from the testing environments. With mixreg (mixing $s$ and $y$ together), the learned value function (bottom) is smoother than the one learned with mixing observations $s$ only.

Specifically, mixreg generates each augmented observation $\tilde{s}$ by convexly combining two observations $s_i, s_j$ randomly drawn from a collection of transitions:

$$\tilde{s} = \lambda s_i + (1 - \lambda) s_j, \tag{5}$$

where $\lambda$ is drawn from a beta distribution $\text{Beta}(\alpha, \alpha)$ with $\alpha \in (0, \infty)$. This is equivalent to sampling a new observation from the convex hull of two distinct observations. Thus, the augmented observation $\tilde{s}$ can smoothly interpolate between the two observations from different training environments, effectively increasing the training data diversity. The supervision signal (e.g. reward) associated with the observation of larger mixing weight is used as the supervision signal for $\tilde{s}$.

However, only mixing the observations following Eq. 5 may not always be effective on lifting the generalization performance of the learned agent (see Section 4.3), possibly because the supervision signal associated with the original observation $s_i$ or $s_j$ is not proper for the interpolated observation $\tilde{s}$. Therefore we further introduce regularization over the supervision signal in a similar interpolation form. Specifically, let $y_i$ denote the associated supervision signal for the state $s_i$, which can be the reward or state value. Mixreg introduces the following mixture regularization:

$$\tilde{y} = \lambda y_i + (1 - \lambda) y_j. \tag{6}$$

As the virtual observation $\tilde{s}$ may largely deviate from $s_i, s_j$, such interpolation would provide a proper supervision signal for $\tilde{s}$. Incorporating $\tilde{s}, \tilde{y}$ would increase the training data diversity and regularize the learning process. From Eqns. (5), (6), mixreg imposes linearity regularization between the observation and corresponding supervision, which helps learn a smooth policy and value function. Figure 2 visualizes different learned value functions. We can see that interpolating the supervision signals indeed helps learn smoother value functions compared to only mixing states. Additional results are provided in the supplementary material.

### 3.3 Application to policy gradient and deep Q-learning

Mixreg can be applied to both policy gradient and deep Q-learning algorithms. In the case of policy gradient, with the interpolated observation and supervision $\tilde{s}, \tilde{y}$, the objective changes from (2) to

$$\tilde{L}^{\text{PG}}(\theta) = \hat{\mathbb{E}}_{i,j} \left[ \log \pi_\theta(\tilde{a}|\tilde{s}) \tilde{A} \right], \tag{7}$$

where $\tilde{A} = \lambda A_i + (1 - \lambda) A_j$, and $\tilde{a}$ is $a_i$ if $\lambda \geq 0.5$ or $a_j$ otherwise.

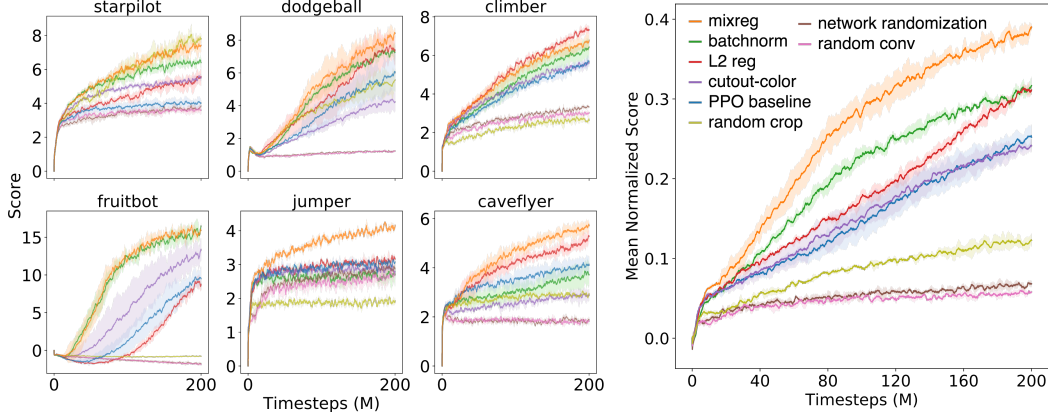

Figure 3: Testing performance of different methods on 500 level generalization. **Left**: Testing performance on individual environments. **Right**: Mean normalized score averaged over all environments.

Similarly, for DQN, the vanilla objective in Eq. (3) becomes

$$\tilde{L}^{\text{DQN}}(\theta) = \hat{\mathbb{E}}_{i,j} \left[ \left( \tilde{R} + \gamma \max_{a'} \tilde{Q}_{\bar{\theta}}(a') - Q_\theta(\tilde{s}, \tilde{a}) \right)^2 \right], \tag{8}$$

where $\tilde{R} = \lambda R_i + (1 - \lambda) R_j$, $\tilde{Q}_{\bar{\theta}}(a') = \lambda Q_{\bar{\theta}}(s'_i, a') + (1 - \lambda) Q_{\bar{\theta}}(s'_j, a')$, and $\tilde{a}$ is $a_i$ if $\lambda \geq 0.5$ or $a_j$ otherwise. To be specific, during the optimization phase of an RL algorithm, we first sample a mixing coefficient $\lambda$ for each collected transition and then mix each transition with another randomly drawn from the same batch. The optimization is performed on the augmented transitions. The above two objectives Eqns. (7), (8) give the basic formulation of applying mixreg in policy-based and value-based RL algorithms. Details about applying mixreg to PPO and Rainbow are given in the supplementary material.

## 4 Experiments

In this section, we aim at answering three questions. 1) Is mixreg able to improve generalization in terms of testing performance in new environments? 2) Is mixreg applicable to different reinforcement learning algorithms and model sizes? 3) How does mixreg take effect for boosting RL generalization? We conduct experiments on the large-scale Procgen Benchmark [2] to answer each one of them.

The Procgen Benchmark presents a suite of 16 procedurally generated game-like environments where visual changes exist between training and testing levels. For most of our experiments, we choose 6 environments (Caveflyer, Climber, Dodgeball, Fruitbot, Jumper, Starpilot) with large generalization gaps. More explanations on such choice are provided in the supplementary material. For RL algorithms, we use Proximal Policy Optimization (PPO) [18] for most experiments considering its strong performance and for fair comparison with previous works. We also use Rainbow [8] to show the applicability of mixreg to value-based RL algorithms. Following [2], we use the convolutional network architecture proposed in IMPALA [5]. Results are averaged over 3 runs and the standard deviations are plotted as shaded areas. For some experiments, we additionally compute the mean normalized scores over different games following [2], in order to report a single score across the 6 environments. Hyperparameters, full training curves and other implementation details are provided in the supplementary material.

### 4.1 Can mixreg improve generalization performance of RL agents?

Following [2], we adopt the "500 level generalization" as our main evaluation protocol. Specifically, an agent is trained on a limited set of 500 levels, and evaluated w.r.t. its zero-shot performance averaged over unseen levels at testing. Unseen levels typically have different background images or different layouts, which are easy for humans to adapt to but challenging for RL agents.

We compare the proposed mixreg with cutout-color, random crop and random convolution that achieve high performance gain in [12], and $\ell_2$ regularization and batch normalization that outperform

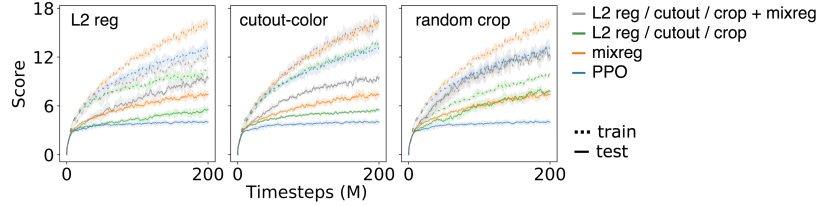

Figure 4: Training and testing performance of combining mixreg with $\ell_2$ regularization, random crop and cutout-color on 500 level generalization, in `Starpilot`.

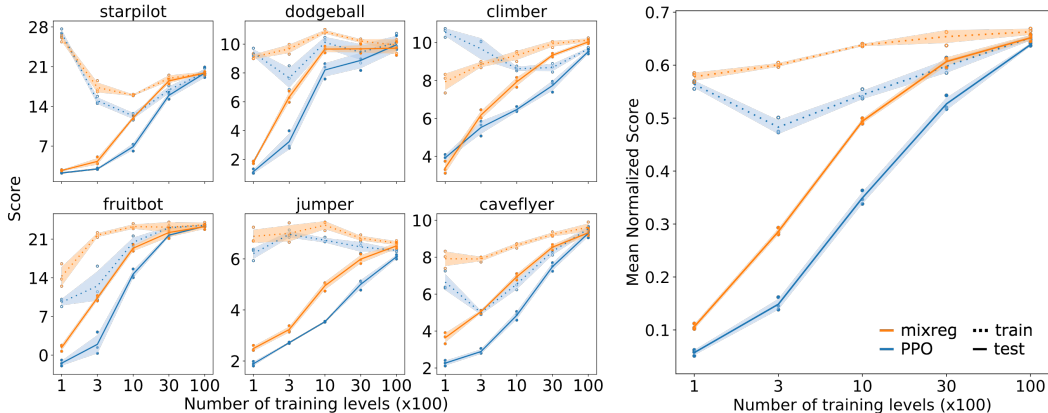

Figure 5: Training and testing performance as a function of the number of training levels. Circles represent individual runs.

other regularization techniques according to [3]. We also include network randomization [13] for comparison, which is the predecessor to random convolution. Results are plotted in Figure 3. We can see mixreg outperforms PPO baseline by a large margin and offers more consistent performance boost than $\ell_2$ regularization and batch normalization. The data augmentation methods (cutout-color, random crop and random convolution) perform worse, in some environments even worse than PPO baseline. This is possibly because data augmentation via local perturbation does not effectively increase the training diversity but creates additional discrepancy between training and testing. We also evaluate mixreg on other 10 environments in Procgen and the results are included in the supplementary material. Moreover, on environments where other augmentation or regularization methods outperform the PPO baseline, such as `Starpilot`, combining our mixreg with them can further improve testing performance and reduce the generalization gap, as shown in Figure 17.

Following [2], we further evaluate generalization performance regarding different numbers of training levels to better validate effectiveness of our method. We plot the results in Figure 5. In comparison with the PPO baseline, our method significantly improves the agents' zero-shot performance in testing environments across different numbers of training levels. Mixreg requires fewer training levels to reach the same testing performance as the PPO baseline, demonstrating its effectiveness in increasing the training data diversity. Moreover, we can observe in Figure 5 that the performance of the PPO baseline often drops first, implying overfitting on the limited training levels. In comparison, our mixreg is less prone to overfitting, showing good regularization effects.

## 4.2 Is mixreg widely applicable?

**Scaling model size** Increasing the model size is shown to significantly improve generalization [2]. A natural question is whether the proposed mixreg brings improvement across different model sizes. Thus we evaluate how mixreg performs with networks of varying sizes. Following [2] we scale the number of convolutional channels at each layer by 2 or 4. The generalization performance on each environment is shown in Figure 6. Across different model sizes, our method exhibits great performance gain compared to the PPO baseline. On environments with little improvement observed in final testing performance, such as `Fruitbot`, mixreg is still more sample-efficient than the baseline.

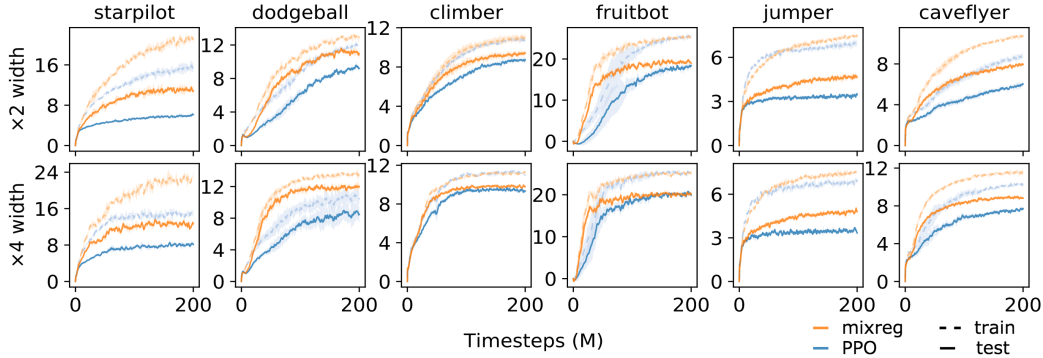

Figure 6: Training and testing performance w.r.t. different model sizes on 500 level generalization.

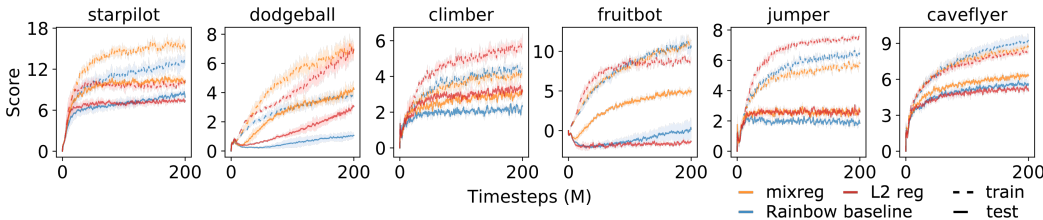

Figure 7: Training and testing performance of Rainbow with mixreg on 500 level generalization.

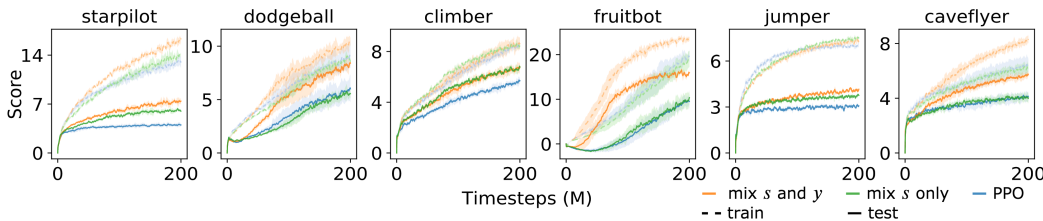

Figure 8: Ablation results of whether interpolating supervision signals.

**Applying to value-based RL algorithm**    In Section 3 we have discussed the application of mixreg to the value-based RL algorithm. Here we conduct experiments to evaluate its effectiveness. Following [2], we use Rainbow [8] and test how much improvement mixreg can bring. We also include a variant using $\ell_2$ regularization for comparison. Results in Figure 7 demonstrate that mixreg is also applicable to the value-based RL algorithm and yields significant improvement on generalization.

### 4.3    How does mixreg benefit generalization performance?

**Benefits of mixing supervision**    When using data augmentation to increase data diversity, a natural choice is to only augment the input observations. However, our proposed mixreg also involves mixing the associated supervision signal (Eqn. (6)). To investigate the benefits of mixing supervision, we conduct an ablation experiment to see how well it performs with only mixing the observations. From the results in Figure 8, we can see in some games only mixing the observations does not bring any performance improvement over the baseline. The mixing of the supervision signals is necessary to effectively increase the training diversity and important for improving policy generalizability.

**Benefiting representation learning**    It is found that applying regularization to DQN helps learn representations more amenable to fine-tuning [6]. To see how the mixreg serves as regularization to improve the learned representation, we conduct two finetuning experiments. First, we fix the feature extraction part in the network and finetune the policy head and value head on the testing levels for 60M timesteps. As shown in Figure 9 (top), when finetuned on testing environments, the policy learned with mixreg achieves much higher performance. Secondly, we finetune the entire model

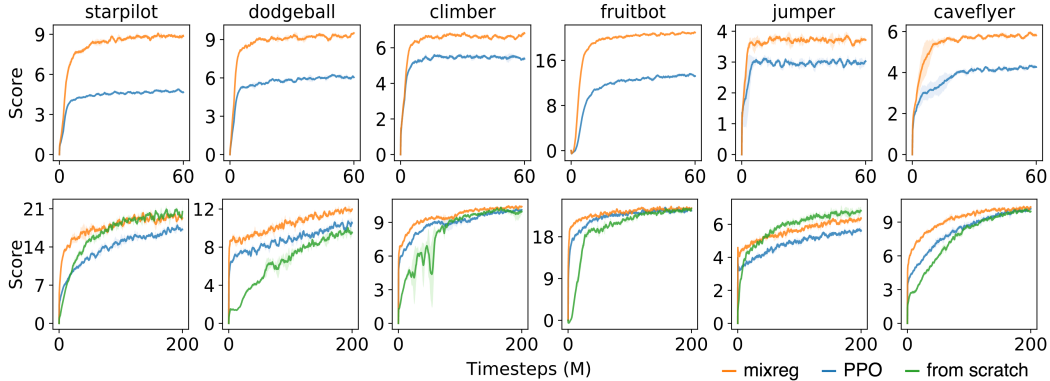

Figure 9: Performance of finetuning trained policies on testing levels with representation fixed (**Top**) or learnable (**Bottom**).

on the testing levels and compare the learned policy to the one trained from scratch. As shown in Figure 9 (bottom), mixreg significantly improves the sample efficiency when the trained policy is finetuned in the new environments. The above results demonstrate that mixreg can help learn more adaptable representation.

# 5   Related works

Generalization in RL draws increasing attention in recent years. Some works [3, 29, 2] show the generalization gap comes from the limited diversity of training environments. This provides a direction to minimize the generalization gap via increasing the training diversity. Cobbe et al. [3] find augmenting the observations with cutout [4] helps improve generalization. Lee et al. [13] propose to use a randomized convolutional layer for randomly perturbing the input observations. Apart from cutout and random convolution, Laskin et al. [12] further evaluate a wide spectrum of data augmentations on improving generalization. Concurrently, Kostrikov et al. [11] propose DrQ to apply data augmentations in model-free RL algorithms. Rather than finding the best augmentation for all tasks, Raileanu et al. [17] propose to automatically select an appropriate augmentation for each given task. In addition to increasing training diversity, regularization techniques such as $\ell_2$ regularization, dropout [24] and batch normalization [10] also help improve RL agents' generalizability [6, 3, 9]. Our work follows the direction of increasing training diversity but proposes a more effective approach than prior data augmentation techniques. It can be combined with other methods such as $\ell_2$ regularization to yield further improvement.

Our work is also related to mixup [32] in supervised learning. Mixup establishes a linear relationship between the interpolation of the features and that of the class labels, increasing the generalization and robustness of the trained classification model. The authors of [32] argued the interpolation principle seems like a reasonable inductive bias that can be possibly extended to RL. Inspired by this, our work investigates on whether enforcing a linear relationship between interpolations of observations and supervision signals helps improve generalization in RL and how it affects the learned policy. To the best of our knowledge, only one existing work [22] applies mixup in RL but their aim is distinct from ours. They uses mixup regularization for learning a reward function while our target is improving generalization.

# 6   Conclusion

In this work, we propose to enhance generalization in RL from the aspect of increasing the training data diversity. We find that existing data augmentation techniques are not suitable as they only perform local perturbation without sufficient diversity. We then consider exploiting the mixture of observations from different training environments. We introduce a simple approach *mixreg* which trains the policy model on the mixture of observations with the correspondingly mixed supervision signals. We demonstrate that our mixreg is much more effective than well-established baselines on

the large-scale Procgen benchmark. Furthermore, we analyze why our mixreg works well and find, besides its effectiveness on increasing data diversity, there are other two contributing factors: mixreg helps learn smooth policies; mixreg helps learn better observation representations. In future, we will explore more flexible and expressive mixing schemes for observations and supervisions. We are also interested in exploring how it will perform in other domains.

## Acknowledgements

Jiashi Feng was partially supported by AISG-100E-2019-035, MOE2017-T2-2-151, CRP20-2017-0006 and NUS_ECRA_FY17_P08.

## Broader Impact

Reinforcement learning has been applied to various domains for learning decision-making agents, including games, intelligent control, robotics, finance and data analytics. Reinforcement learning tends to suffer poor generalization performance when the trained agent is deployed in a new environment. This work proposes a simple data augmentation based solution that substantially improves RL agents' generalization performance. This work would have following positive influences in this field. The proposed method is simple and easy to deploy, and would improve generalization performance and robustness of RL agents in various environments. This will be inspiring for following research works, and also benefit deployment of RL agents in practice and help develop trustworthy agents. On the flip-over side, the effectiveness of the proposed method is only verified in the game domain. It remains unclear how it will perform in other domains like finance, where the data have different modalities. Improper deployment of the proposed method may even worsen the performance of the RL agents trained with the proposed method and hurt the quality of output decisions.

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
