[Supplementary Material]

## Appendix A  Application of mixreg to PPO and Rainbow

### A.1  PPO

**Background**  Proximal Policy Optimization (PPO) [12] introduces a novel objective function, resulting in a policy gradient method which is simpler, more general and performs better than previous work Trust Region Policy Optimization (TRPO) [10]. In each update, the algorithm collects a batch of transition samples using a rollout policy $\pi_{\theta_{old}}(a_t|s_t)$ and maximizes a clipped surrogate objective:

$$L^{\text{CLIP}}(\theta) = \hat{\mathbb{E}}_t \left[ \min(r_t(\theta)A_t, \text{clip}(r_t(\theta), 1 - \epsilon, 1 + \epsilon)A_t) \right], \tag{9}$$

where $r_t(\theta)$ denotes the probability ratio $r_t(\theta) = \frac{\pi_\theta(a_t|s_t)}{\pi_{\theta_{old}}(a_t|s_t)}$, $A_t$ is the advantage at timestep $t$ estimated by generalized advantage estimation (GAE) [11] and $\epsilon$ is a hyperparameter controlling the width of the clipping interval. GAE makes use of a learned state-value function $V(s)$ for computing the advantage estimation. To learn the value function, the following loss is adopted:

$$L^V(\theta) = \hat{\mathbb{E}}_t \left[ \frac{1}{2} \max \left( (V_\theta - V_t^{\text{targ}})^2, (\text{clip}(V_\theta, V_{\theta_{old}} - \epsilon, V_{\theta_{old}} + \epsilon) - V_t^{\text{targ}})^2 \right) \right], \tag{10}$$

where $V_t^{\text{targ}}$ is the bootstrapped value function target. Besides, an entropy bonus is often added to ensure sufficient exploration:

$$L^H(\theta) = \hat{\mathbb{E}}_t \left[ H[\pi_\theta](s_t) \right]. \tag{11}$$

Putting the above losses together, the overall minimization objective of PPO is:

$$L^{\text{PPO}}(\theta) = -L^{\text{CLIP}}(\theta) + \lambda_V L^V(\theta) - \lambda_H L^H(\theta), \tag{12}$$

where $\lambda_V, \lambda_H$ are the coefficients to adjust the relative importance of each component. The algorithm alternates between sampling trajectory data using the policy and performing optimization on the collected data based on the above loss.

**Applying mixreg**  At each update, we randomly draw two transitions $i, j$ from the collected batch of transitions and convexly combine the observations and the associated supervision signals as follows:

$$\begin{aligned}
\tilde{s} &= \lambda s_i + (1 - \lambda)s_j, \\
\tilde{\pi}_{\theta_{old}} &= \lambda \pi_{\theta_{old}}(a_i|s_i) + (1 - \lambda)\pi_{\theta_{old}}(a_j|s_j), \\
\tilde{V}_{\theta_{old}} &= \lambda V_{\theta_{old}}(s_i) + (1 - \lambda)V_{\theta_{old}}(s_j), \\
\tilde{V}^{\text{targ}} &= \lambda V_i^{\text{targ}} + (1 - \lambda)V_j^{\text{targ}}, \\
\tilde{A} &= \lambda A_i + (1 - \lambda)A_j.
\end{aligned} \tag{13}$$

Since we deal with discrete actions, the interpolated action $\tilde{a}$ is simply $a_i$ if $\lambda \geq 0.5$ or $a_j$ otherwise. The optimization is performed on the generated batch of mixed transitions and each part of the overall objective in Eqn. (12) now becomes

$$\tilde{L}^{\text{CLIP}}(\theta) = \mathbb{E}_{i,j} \left[ \min(\tilde{r}(\theta)\tilde{A}, \text{clip}(\tilde{r}(\theta), 1 - \epsilon, 1 + \epsilon)\tilde{A}) \right] \quad \text{where } \tilde{r}(\theta) = \frac{\pi_\theta(\tilde{a}|\tilde{s})}{\tilde{\pi}_{\theta_{old}}}, \tag{14}$$

$$\tilde{L}^V(\theta) = \mathbb{E}_{i,j} \left[ \frac{1}{2} \max \left( (V_\theta(\tilde{s}) - \tilde{V}^{\text{targ}})^2, (\text{clip}(V_\theta(\tilde{s}), \tilde{V}_{\theta_{old}} - \epsilon, \tilde{V}_{\theta_{old}} + \epsilon) - \tilde{V}^{\text{targ}})^2 \right) \right], \tag{15}$$

$$\tilde{L}^H(\theta) = \mathbb{E}_{i,j} \left[ H[\pi_\theta](\tilde{s}) \right]. \tag{16}$$

### A.2  Rainbow

**Background**  Rainbow [6] combines the following six extensions for the DQN algorithm: double DQN [14], prioritized replay [9], dueling networks [15], multi-step learning [13], distributional RL [1] and Noisy Nets [5]. As in distributional RL [1], Rainbow learns to approximate the distribution of returns instead of the expected return. At each update, the algorithm samples a batch of transitions from the replay buffer and minimizes the following Kullback-Leibler divergence between the predicted distribution and the target distribution of returns:

$$\hat{\mathbb{E}}_t \left[ D_{\text{KL}}(\Phi_{\boldsymbol{z}} d_t^{(n)} \| d_t) \right]. \tag{17}$$

Here $d_t$ denotes the predicted distribution with discrete support $\boldsymbol{z}$ and probability masses $\boldsymbol{p}_\theta(s_t, a_t)$ and $d_t^{(n)}$ denotes the target distribution with discrete support $R_t^{(n)} + \gamma_t^{(n)}\boldsymbol{z}$ and probability masses $\boldsymbol{p}_{\bar{\theta}}(s_{t+n}, a_{t+n}^*)$, where $a_{t+n}^*$ denotes the bootstrap action. The target distribution is constructed by contracting the value distribution in $s_{t+n}$ according to the cumulative discount $\gamma_t^{(n)}$ and shifting it by the truncated $n$-step discounted return $R_t^{(n)} \equiv \sum_{k=0}^{n-1} \gamma_t^{(k)} R_{t+k+1}$, where $\gamma_t^k = \prod_{i=1}^{k} \gamma_{t+i}$ with $\gamma_t = \gamma$ except on episode termination where $\gamma_t = 0$. $\Phi$ is a L2-projection of the target distribution onto the support $\boldsymbol{z}$. The bootstrap action $a_{t+n}^*$ is greedily selected by the *online network* and evaluated by the *target network*. Following prioritized replay [9], Rainbow prioritizes transitions by the KL loss. Rainbow uses a dueling network architecture adapted for use with return distributions. Following Noisy Nets [5], all linear layers are replaced with their noisy equivalent.

**Applying mixreg**   Similarly, at each update, we randomly draw two transitions $i, j$ from the sampled batch of transitions and interpolate the observations and the associated supervision signals:

$$
\begin{aligned}
\tilde{s} &= \lambda s_i + (1-\lambda)s_j, \\
\tilde{\boldsymbol{p}}_{\bar{\theta}} &= \lambda \boldsymbol{p}_{\bar{\theta}}(s_{i+n}, \tilde{a}^*) + (1-\lambda)\boldsymbol{p}_{\bar{\theta}}(s_{j+n}, \tilde{a}^*), \\
\tilde{R}^{(n)} &= \lambda R_i^{(n)} + (1-\lambda)R_j^{(n)},
\end{aligned}
\tag{18}
$$

where $\tilde{a}^*$ is $a_{i+n}^*$ if $\lambda \geq 0.5$ or $a_{j+n}^*$ otherwise. Note that mixing $\boldsymbol{p}_{\bar{\theta}}$ corresponds to mixing Q-value mentioned in Section 3.3 since $Q_{\bar{\theta}}(s, a) = \boldsymbol{z}^\top \boldsymbol{p}_\theta(s, a)$. The optimization objective in Eqn. (17) now becomes

$$
\hat{\mathbb{E}}_t \left[ D_{\mathrm{KL}}(\Phi_{\boldsymbol{z}} \tilde{d}_t^{(n)} \| \tilde{d}_t) \right].
\tag{19}
$$

Here $\tilde{d}_t$ is the new predicted distribution with probability masses $\boldsymbol{p}_\theta(\tilde{s}, \tilde{a})$ where $\tilde{a}$ is $a_i$ if $\lambda \geq 0.5$ or $a_j$ otherwise, and $\tilde{d}_t^{(n)}$ is the new target distribution with discrete support $\tilde{R}^{(n)} + \tilde{\gamma}^{(n)}\boldsymbol{z}$ and probability masses $\tilde{\boldsymbol{p}}_{\bar{\theta}}$ where $\tilde{\gamma}^{(n)}$ is $\gamma_i^{(n)}$ if $\lambda \geq 0.5$ or $\gamma_j^{(n)}$ otherwise.

## Appendix B   On the smoothness of the learned policy and value function

In this part, we provide additional results to demonstrate that mixreg helps learn a smooth policy and value function. Figure 11 plots the learned value function in Figure 2 in 3D space for better illustration. The color map is slightly different from the one used in Figure 2 but this does not affect the result. Moreover, we compute the empirical Lipschitz constant [17] of the trained network, i.e. calculating the following ratio for each pair of observations $s_i$, $s_j$ and taking the maximum:

$$
\frac{\|f(s_i) - f(s_j)\|}{\|s_i - s_j\|}
\tag{20}
$$

where $f(s)$ denotes the latent representation of observation $s$. Specifically, We collect a batch of observations and sample $10^6$ pairs for estimating the empirical Lipschitz constant. The results are aggregated in box plots, shown in Figure 10. We can see that the network trained with mixreg has smaller empirical Lipschitz constant compared to the PPO baseline, implying that mixreg helps learn smoother policy.

Figure 10: The distribution of the calculated ratios from $10^6$ randomly sampled pairs of observations. The largest ratio corresponds to the estimated empirical Lipschitz constant.

Figure 11: 3D visualization of the learned state-value functions $V(s)$ in PPO for 6 game environments.

# Appendix C  Implementation details and hyperparameters

Due to high computation cost, we choose 6 out of 16 environments from the Procgen benchmark. Among the 16 environments, we first exclude 3 environments (Chaser, Leaper, Bossfight) which do not exhibit large generalization gap under 500 level generalization protocol. We then exclude two difficult games (Maze and Heist) where we find the trained policy by PPO performs comparably to a random policy. In the remaining 11 games, we randomly choose 6 environments (Caveflyer, Climber, Dodgeball, Fruitbot, Jumper, Starpilot) for evaluating different methods.

Following [2], we use the convolutional architecture proposed in IMPALA [4]. When applying batch normalization, we add a batch normalization layer after each convolution layer following the implementation[1] in [3]. When applying $\ell_2$ regularization, we use a weight $10^{-4}$ as suggested by [3]. For cutout-color and random convolution, we follow the official implementation[2] in [7]. As the official implementation of random crop is problematic[3] (cropping $64 \times 64$ window out of $64 \times 64$ observation), we implement our own version of random crop by first resizing observations to $75 \times 75$ and then randomly cropping with a $64 \times 64$ window. For network randomization method [8], we follow their implementation[4] but do not adopt the Monte Carlo approximation with multiple samples during inference.

For both PPO and Rainbow experiments, we use the same hyperparameters as in [2] except for the ones with † in the following tables. For PPO experiments, we halve the number of workers but double the number of environments per worker to fit our hardware. This should result in little difference on performance and we are able to reproduce the results in [2]. For Rainbow experiments, as the implementation in [2] is not available, we follow the implementation in anyrl-py[5] and some hyperparameters (denoted with †) in retro-baselines[6]. $R_{max}$ is the normalization constant used in [2] and the distributional min value is changed to -5 in FruitBot.

| PPO | |
|---|---|
| Env. distribution mode | Hard |
| $\gamma$ | .999 |
| $\lambda$ | .95 |
| # timesteps per rollout | 256 |
| Epochs per rollout | 3 |
| # minibatches per epoch | 8 |
| Entropy bonus coefficient ($\lambda_H$) | .01 |
| Value loss coefficient ($\lambda_V$) | .5 |
| Gradient clipping ($\ell_2$ norm) | .5 |
| PPO clip range | .2 |
| Reward normalization? | Yes |
| Learning rate | $5 \times 10^{-4}$ |
| †# workers | 2 |
| †# environments per worker | 128 |
| Total timesteps | 200M |
| LSTM? | No |
| Frame stack? | No |
| beta distribution parameter $\alpha$ | 0.2 |

| Rainbow | |
|---|---|
| Env. distribution mode | Hard |
| $\gamma$ | .999 |
| Learning rate | $2.5 \times 10^{-4}$ |
| # workers | 8 |
| # environments per worker | 64 |
| # env. steps per update per worker | 64 |
| Batch size per worker | 512 |
| Reward clipping? | No |
| Distributional min/max values | $[0, R_{\max}]$ |
| †Memory size | 500K |
| †Min history to start learning | 20K |
| Exploration $\epsilon$ | 0.0 |
| Noisy Nets $\sigma_0$ | 0.5 |
| †Target network period | 8192 |
| Adam $\epsilon$ | $1.5 \times 10^{-4}$ |
| Prioritization exponent $\omega$ | 0.5 |
| †Prioritization importance sampling $\beta$ | 0.4 |
| Multi-step returns $n$ | 3 |
| Distributional atoms | 51 |
| Total timesteps | 200M |
| LSTM? | No |
| Frame stack? | No |
| beta distribution parameter $\alpha$ | 0.2 |

# Appendix D  Ablation results of varying the beta distribution parameter $\alpha$

For the beta distribution $\text{Beta}(\alpha, \alpha)$ used to draw mixing coefficient $\lambda$, we choose $\alpha = 0.2$ from the interval $[0.1, 0.4]$ suggested by [16]. We also test different $\alpha$ and plot the results in Figure 12. Using too large $\alpha$ (e.g. 1.0) leads to performance degradation in certain environments.

Figure 12: Training and testing performance of mixreg with different parameter $\alpha$ for $\text{Beta}(\alpha, \alpha)$.

# Appendix E   Additional results and training curves on Procgen

## E.1   Comparing different methods on 500 level generalization in 6 environments

Figure 13: Training and testing performance of different methods on 500 level generalization.

Figure 14: Mean normalized score of different methods on 500 level generalization.

Figure 15: Training and testing performance on 500 level generalization in 16 environments.

Figure 16: Mean normalized score on 500 level generalization averaged over 16 environments.

## E.3  Combining mixreg with other methods on 500 level generalization

Figure 17: Training and testing performance of combining mixreg and other methods on 500 level generalization.

## E.4  Scaling model size

Figure 18: Mean normalized score w.r.t. different model sizes on 500 level generalization.

## E.5 Varying the number of training levels

Figure 19: Training and testing performance w.r.t. different number of training levels.

Figure 20: Mean normalized score w.r.t. different number of training levels.

## Footnotes

[1]https://github.com/openai/coinrun

[2]https://github.com/pokaxpoka/rad_procgen

[3]https://github.com/pokaxpoka/rad_procgen/issues/1

[4]https://github.com/pokaxpoka/netrand

[5]https://github.com/unixpickle/anyrl-py

[6]https://github.com/openai/retro-baselines