[Reviews · NeurIPS 2020]

Review 1

Summary and Contributions: The paper aims to improve generalization in RL, defined as performance of an RL agent on test-set environments it has not gotten data from at training time. To do so, they take inspiration from mixup regularization, and propose mixreg regularization. Random observations are linearly combined according to a lambda sampled from a Beta distribution, with their targets linearly interpolated as well, to encourage estimates of Q, A, R, and \pi to be more linear. (One important detail is that Q/A/R are interpolated, but interpolation between a_i, a_j is binary based on whether lambda >= 0.5 or not.) Evaluated on ProcGen using PPO and Rainbow, they find it does better than a number of other proposed data augmentation and regularization methods.

Strengths: The evaluations on ProcGen are fairly extensive, and mixreg is very definitively better than no regularization on PPO. Mixreg also has some improved benefit when combined with methods like L2 reguarlization. Visualizations of the function landscape qualitatively back up the claim that mixreg encourages the model to be more linear when generalizing between points. The paper hits the spot of "intuitively a good idea, yet somehow wasn't done in the literature yet", based on a search I did for related papers, and I believe the overall work is good.

Weaknesses: The paper cites the RAD paper, comparing their method to cutout and random convolutions. However, the RAD paper found these were both much weaker than a random crop of the image, and their final results only use random cropping. I therefore found it strange why random cropping was not baselined against. This feels like the biggest strike against the paper. The paper includes a mixed observation baseline, where just the states are blended, without changing the targets. The authors argues this increases input diversity, but immediately acknowledge that the supervision signal for s_1 or s_2 may not match for \tilde{s}. I would go further and say it *usually* does not match, since mixing observations is very unlikely to keep the optimal Q-value the same. Because of this, I don't understand why the mixed observation baseline was included at all. Any conclusions that could be made about the increased input diversity are cancelled out by the ill-formed optimization problem.

Correctness: The core method looks correct, but I have some question on the details. In the entropy term in PPO, it appears the computation uses the entropy of the current policy, rather than the mixed policy. This seems like it was done because the entropy of the mixed policy is hard to compute in closed form (entropy of mixed policy requires knowing every pair of states s_1, s_2 that could be mixed into a single \tilde{s}), but this doesn't seem to be discussed. Eqn (14) in the appendix should be pi_old(a|s) in the denominator. Eqn (18) should not have tildes for the right hand side of the R equation.

Clarity: The paper is clear about its proposed method and its evaluation methodology.

Relation to Prior Work: I had assumed someone had applied mixup to RL already, but had trouble finding prior work, so this does look like a new contribution. The paper does appropriately cite prior generalization work and the mixup paper. I did find the paper "End-to-End Robotic Reinforcement Learning without Reward Engineering", which uses mixup regularization when learning a reward function. I believe it is distinct, but should be cited.

Reproducibility: Yes

Additional Feedback: Although I believe the arguments for mixup style regularization make sense, I do have some concerns about potential bias from the ProcGen benchmark. Many of the games in ProcGen are 2D games with a fixed camera (a skim of videos in the envs gives 8 of 16 envs have a fixed cameras and 7 of those 8 have a static image background.) We would expect a mixup style method to do better on these environments, because averaging 2 images together naturally exposes what parts of the image are static, and what parts of the image are not. (Parts that move will be fuzzier in the average, parts that do not move will be sharp.) So I have some concerns over how well this will generalize to other settings. Based on the training curves, mixup is simply more efficient than PPO on the train-time environments. To me this suggests that it may not be creating better generalization behavior - it may just be a better RL optimization, which would be a worthy paper in its right. It was not clear to me whether mixreg was applied during finetuning of policies in the test environment (Figure 9). It seems strange that mixreg is more efficient than plain PPO, given that regularization tends to increase training time due to the increase in data complexity. The authors say picking too large alpha for Beta(\alpha, \alpha) distribution hurts performance, but performance between different \alpha choices appears fairly similar. Mixreg is combined with L2 regularization, but it did not look like it was combined with cutout color, random conv, etc. Why? Edit: i have read the other reviews and rebuttal. I still feel this work is valuable, but do want to push back on the authors to consider the following: * Although the ProcGen results are encouraging, I would still like to see experiments in another benchmark set, to be sure it is not a ProcGen specific conclusions. * It is true that mixup is more of a data augmentation technique, and it can be compared against regularization methods, but it is also important to compare how it combines with other data augmentation techniques. Some augmentations combine well with other augmentations, and some do not, and it is important to know which. * Even if the exact implementation of crop used is unclear, I feel it is still valuable to compare against your own implementation of cropping based on what the paper describes. It may not exactly match their code, but if their code is unclear, then implementing from the paper description should be done instead. The cropping is either done by padding the image and cropping from the padded image, or generating a larger image by resizing and cropping the larger one.


Review 2

Summary and Contributions: The authors propose a simple regularization "mixreg", heavily inspired by the "mixup" method from supervised learning. The idea is to create new data, not by perturbing one input (here images), but by convexly mixing two inputs and their outputs. While the idea is simple, it is thoroughly evaluated on the ProcGen benchmark, including some additional ablation studies. Furthermore, the results indicate that it provides a clear advantage on some games, as well as on the overall average.

Strengths: - Strong experimental results - The topic of generalization in RL is important - Simple algorithm, little computational overhead

Weaknesses: - The algorithm is very similar to the existing "mixup" algorithm, which, however, wasn't applied to RL yet

Correctness: Yes.

Clarity: Yes.

Relation to Prior Work: Yes.

Reproducibility: Yes

Additional Feedback: I find this paper tricky to evaluate. It is well written with thorough experiments (requiring a lot of compute) and strong results and as such valuable to the community. On the other hand, it is a very straightforward application of a regularization algorithm from supervised learning to RL (with some minor tweak to adapt it). One could argue that this is part of the novelty, to realize that existing methods can also be applied to RL, where they haven't been yet tested. On the other hand, one could argue that re-using existing methods and "only" thoroughly evaluating them using significant computing resources should not be a NeurIPS paper. I currently believe, given the relative rarity of similar "apply existing methods to RL" papers, and the excellent execution of the paper, that it is a valuable contribution to NeurIPS and should be accepted. However, I am also looking forward to input from the authors on those questions, as well as an active discussion with other reviewers (assuming some of them will disagree with me). [After author response:] Thank you for your response! As I said in my original review I find this paper difficult to evaluate. I find it valuable due to its simplicity and positive results, however, as in particular R3 pointed out, some additional comparisons or insights would make this a much stronger paper. As such, I still vote for acceptance, however, I decided to reduce my score 7 -> 6.


Review 3

Summary and Contributions: An application of mixup in supervised learning to improve generalization in RL to visual changes.

Strengths: -- A simple approach to improve generalization in RL on visual benchmarks such as ProcGen which works better than naive data augmentation. -- Mixup (or mixreg) may serve as an important baseline for ProcGen as well as seems useful for practical RL methods.

Weaknesses: -- Lack of novelty: I found the work to be extremely incremental and straightforward application of mixup. The main finding is that mixup performs reasonably well and outperforms simple data augmentation baselines proposed recently. Mixup can easily be applied to regression/classification problems and policy/value based methods are simply examples of such problems. Overall, the paper is too incremental for NeurIPS. -- Lack of new insights: The paper doesn't present any new insights but empirically shows that mixup helps to improve generalization of neural networks trained in RL, an unsurprising finding. -- Lack of theoretical grounding

Correctness: Yes.

Clarity: Yes, the paper is easy to understand.

Relation to Prior Work: Yes, however the paper misses a couple of recent (unpublished) data augmentation papers: 1. Kostrikov, I., Yarats, D., & Fergus, R. (2020). Image augmentation is all you need: Regularizing deep reinforcement learning from pixels. arXiv preprint arXiv:2004.13649. 2. Raileanu, R., Goldstein, M., Yarats, D., Kostrikov, I., & Fergus, R. (2020). Automatic Data Augmentation for Generalization in Deep Reinforcement Learning. arXiv preprint arXiv:2006.12862.

Reproducibility: Yes

Additional Feedback: Post Rebuttal: After reading the rebuttal, I have updated my score since I may have been overly harsh in reviewing the contribution. However, I still feel that the paper requires additional analysis for a solid contribution and just empirically evaluating an existing algorithm and presenting SOTA results on a single benchmark seems limited for a NeurIPS paper. --------------------------------- More analysis is needed (such as section 4.3) -- understanding why mixreg works as well would likely help in improving the paper. Some suggestions: -- Can the increase in training diversity be characterized more rigorously? -- How the network's representation changes when trained with and without mixup? -- Is it possible to characterize the generalization benefit in terms of the increased number of MDPs (created due to mixup) and how does this improve upon naive data augmentation? -- It is unclear when would would mixup fail in RL (for example, would this fail when we mix two extremely different environments (let's say Atari and Procgen) -- mixing cat and dog images and their labels does seem to work in supervised learning) Overall, the paper seems to be quite incremental and I am not sure if the above improvements would be enough for a strong contribution.


Review 4

Summary and Contributions: This paper proposes a new data augmentation algorithm mixreg for generalization in deep RL. It is based on the mixup algorithm from supervised learning, creating new data points by interpolating between seen state observations and their corresponding value functions. Experiments on the ProcGen benchmark show improvements over existing data augmentation and regularization methods.

Strengths: The proposed algorithm mixreg is intuitively simple. When combined with PPO, it leads to substantial improvements over previous data augmentation and regularization methods in deep RL, so I believe it would be of interest to the community.

Weaknesses: The mixreg algorithm is a straightforward extension of the mixup algorithm from supervised learning, although I do not consider this to be a major weakness. It would be good to provide some theoretical backing for the intuition that mixreg leads to a smoother policy and better generalization performance mentioned in lines 41-43. In addition, figure 7 compares mixreg only to Rainbow. For a complete empirical evaluation based on the claims made in the introduction, I would expect comparisons to other augmentation/regularization techniques for value-based RL algorithms as well. In section 4.2, it would be good to test on a wider range of architectures in order to convincingly argue that mixreg scales to different model sizes.

Correctness: This paper is a purely empirical work. The empirical methodology seems reasonable, although I did not go through the code in detail.

Clarity: The paper is clearly written, easy to understand. In particular, I found figure 2 to be illuminating.

Relation to Prior Work: The paper provides literature review of previous approaches for generalization in RL in game-like environments (e.g. ProcGen) and compares the proposed algorithm to previous work. In my opinion this is sufficient.

Reproducibility: Yes

Additional Feedback: # Update After reading the other reviews and the author response, I do not plan to change my score. The authors have partially addressed my concerns, in particular about Rainbow.

[Author Response · NeurIPS 2020]

**To R1**: ▸ Originally we selected `crop`, `cutout-color` and `random conv` from RAD paper based on their Table 2 and Figure 3b. However, we found their released code of `crop` is confusing (cropping 64x64 window out of 64x64 observation; with unsolved github issue). Thus, we exclude `crop` from experiments. ▸ We agree with "it *usually* does not match". But mixing-obs baseline is similar to data augmentation where input $s$ is perturbed and supervision is from actual $s$. We thus include it for completeness. ▸ The entropy term in plain PPO is calculated using current policy instead of $\pi_{old}$; in mixreg we replace $s$ with the mixed $\tilde{s}$ for entropy calculation. We will correct Eqn (14), (18) and cite the suggested paper. ▸ We agree that mixing "exposes static parts" but we found mixreg favors games with dynamic backgrounds (see Fig. 15). This is worth future investigation. ▸ In the finetuning stage, mixreg is not applied. We use plain PPO to finetune policies trained with different methods (plain PPO, mixreg). ▸ From Fig. 12, when $\alpha = 1$, the performance drop is noticeable in some games (e.g. `starpilot`, `climber`, `fruitbot`). ▸ Mixreg is more like data augmentation methods, so we combine it with regularization techniques (e.g. L2 regularization) instead of other augmentations.

**To R2**: ▸ Thanks for recognizing our contribution. Although our method is simple, the empirical results are surprisingly good. Besides, it also gives some intriguing observations: (1) it seems to make little sense to mix rewards or Q-values from different environments but the performance is good; (2) despite being discussed in the original mixup paper, applying mixup to RL has been overlooked since, even in three recent papers [1, 2, 3] about using data augmentation in RL. Therefore, we believe our work can inspire new insights into the important topic of generalization in RL.

**To R3**: ▸ Though it is straightforward, applying mixup to RL has been overlooked, even in three recent papers [1, 2, 3] about using data augmentation in RL. Besides, we do not think our findings are trivial. Other reviewers also agree that our work "hits the spot of intuitively a good idea, yet somehow wasn't done in the literature yet" (R1), and "would be a valuable contribution and of interest to the community" (R2 & R4). ▸ We disagree with the reviewer's remark that policy / value based methods are simply examples of regression / classification problems. It diminishes the great progress in the whole RL field. ▸ We will add two prior works mentioned. But the second one is not publicly available before submission deadline. ▸ We thank the reviewer for providing additional feedback on better understanding how mixreg works. We will definitely further pursue along this direction. But lacking some theoretical analysis does not diminish the value of our work. Characterizing the increased training diversity (or the increased number of MDP) is a good direction for future investigation. To analyze changes in the network's representation, we have tried to visualize the hidden features using t-SNE but did not observe meaningful explanation, so we only include the quantitative results of finetuning experiments. We will do further representation analysis. Regarding failure cases of our method, we observe that mixreg struggles in maze-like environments (e.g. `maze`, `miner`) and environments where object color contains important information (e.g. `plunder` where the agent controls a ship to destroy enemy ships marked by different colors while avoid hitting friendly ships marked by same colors).

**To R4**: ▸ As demonstrated in Appendix B, we find mixreg helps decrease the Lipschitz constant of the learned network, which coincides with analysis on mixup in supervised learning context [4]. Smaller Lipschitz constant may lead to smoother policy and better generalization, though deeper reason on why mixing improving generalization is still unclear and worth further exploration. ▸ We conduct additional experiments for Rainbow, but due to limited time for rebuttal, we only manage to finish evaluating Rainbow with L2 regularization on 4 environments (see right figure). Our mixreg is on par with or outperforms L2 regularization for Rainbow. We will include the complete results in the final version. ▸ For evaluating the scalability to different model sizes, we choose multiplying the number of convolutional channels by 2 and 4 for a fair comparison with the baseline in Procgen benchmark.

### Reference

[1] Kostrikov, Ilya, Denis Yarats, and Rob Fergus. "Image augmentation is all you need: Regularizing deep reinforcement learning from pixels." *arXiv preprint arXiv:2004.13649* (2020).

[2] Laskin, Michael, et al. "Reinforcement Learning with Augmented Data." *arXiv preprint arXiv:2004.14990* (2020).

[3] Raileanu, Roberta, et al. "Automatic Data Augmentation for Generalization in Deep Reinforcement Learning." *arXiv preprint arXiv:2006.12862* (2020).

[4] Carratino, Luigi, et al. "On Mixup Regularization." *arXiv preprint arXiv:2006.06049* (2020).


[Meta-Review · NeurIPS 2020]

This submission was generally understood by reviewers to be a straightforward extension of existing work on supervised learning regularization, thus presenting limited technical novelty. It was reasonably well executed from an experimental perspective and potentially high impact given the strength of the results. In discussion, reviewers debated the merits of the paper, with several arguing that for such a limited algorithmic contribution the analysis component needed to be stronger. R3 would have liked to see broader empirical assessment, a greater discussion and interrogation of limitations, and whether combination with other forms of data augmentation yielded additive gains, while R1 felt that evaluation on strictly image-based environments was potentially misleading. I concur with several of these criticisms, but must balance the paper's shortcomings with the value to the community in highlighting a method which is a very clear target for further research, and an already potentially useful entry in a practitioner's toolbox.